# Isotopologue-induced structural dynamics of a triazolate metal-organic framework for efficient hydrogen isotope separation

Linda Zhang [1,2] ✉, Richard Röß-Ohlenroth[3], Vanessa K. Peterson [4], Samuel G. Duyker [5], Cheng Li [6], Jhonatan Luiz Fiorio[7], Jan-Ole Joswig [7], Robert Dinnebier[8], Dirk Volkmer [3] & Michael Hirscher [2,9] ✉

Efficient hydrogen isotope separation remains the biggest challenge due to the nearly identical physicochemical properties of $H_2$ and $D_2$. Through in situ neutron powder diffraction and gas adsorption experiments, we investigate the hydrogen isotopologue-induced structural dynamics of the triazole-based metal-organic framework [Mn(ta)$_2$]. Gas loading induces a measurable lattice expansion, more pronounced for $H_2$ than $D_2$, and two distinct adsorption sites are identified with a subtle but significant difference in the occupancy of $H_2$ and $D_2$ at 60 K. Cryogenic thermal desorption spectroscopy after exposure to a 1:1 isotope mixture reveals an exceptionally high $D_2/H_2$ selectivity of 32.5 at 60 K. When exposed to a $D_2/H_2$ mixture of 5:95, $D_2$ enriches to 75% in a single cycle. Given the commercial availability of the ligand and the scalability of the dia-framework topology across divalent transition metals, upscaling for industrial-scale deuterium separation is a realistic prospect. Our results give crucial molecular-level insights into isotopologue-induced structural dynamics in triazolate-based MOFs and provide guidance for improvement of isotope separation materials.

The global shift towards renewable and clean energy sources has intensified the importance of hydrogen (H) and its isotopes, deuterium (D) and tritium (T)[1]. However, separating these isotopes remains a significant challenge due to their similar physicochemical properties[2]. Recently, porous materials, particularly metal-organic frameworks (MOFs), have garnered attention for their potential in gas separation[3].

MOFs are ordered three-dimensional structures comprising metal ions and organic linkers[4]. They offer an energy-efficient alternative to traditional separation methods, potentially reducing consumption by up to 90%[5]. For hydrogen isotopes, MOFs facilitate separation through two quantum sieving mechanisms: kinetic quantum sieving (KQS) and chemical affinity quantum sieving (CAQS)[6]. KQS occurs in MOFs with small pore apertures, where deuterium's faster diffusion relative to hydrogen under cryogenic conditions is attributed to its shorter de Broglie wavelength[7,8]. CAQS, on the other hand, occurs at strong adsorption sites within the MOFs, where heavier isotopes exhibit stronger interactions due to lower zero-point energy (ZPE)[9,10]. This has led to promising developments in hydrogen isotope separation using MOFs, achieved by finely tuning the pore size[11,12], binding strength[13,14], and framework flexibility[15], which include phenomena such as breathing transitions and gating effects[16].

[1]Frontier Research Institute for Interdisciplinary Sciences, Tohoku University, Sendai 980-0845, Japan. [2]Advanced Institute for Materials Research (WPI-AIMR), Tohoku University, Sendai 980–8577, Japan. [3]Chair of Solid State and Materials Chemistry, Institute of Physics, University of Augsburg, 86159 Augsburg, Germany. [4]Australian Centre for Neutron Scattering, Australian Nuclear Science and Technology Organisation, Lucas Heights, NSW 2234, Australia. [5]Sydney Analytical, The University of Sydney, Sydney, NSW 2006, Australia. [6]Spallation Neutron Source, Oak Ridge National Laboratory, Oak Ridge, TN 37830, USA. [7]Theoretische Chemie, Technische Universität Dresden, 01069 Dresden, Germany. [8]Max Planck Institute for Solid State Research, 70569 Stuttgart, Germany. [9]Max Planck Institute for Intelligent Systems, 70569 Stuttgart, Germany. ✉e-mail: linda.zhang.a3@tohoku.ac.jp; hirscher@is.mpg.de

Growing attention is directed towards the strategic development of triazolate-based frameworks with targeted pore size, particularly given their structural tuneability[17,18]. The three-dimensional, 4-connected dia-type coordination framework $[M(ta)_2]$ (where M represents $Mn^{II}$, $Fe^{II}$, $Co^{II}$, $Cu^{II}$, $Zn^{II}$, $Cd^{II}$, and $Cr^{II/III}$), is constructed using 1H-1,2,3-triazole (H-ta) ligands and metal ions[19,20]. This structure embeds a dia topology pore system, comprising small cavities and narrower connecting channels. Extra pore-surface pockets formed by six triazole ligands connect to these channels with smaller entrance diameters[21,22]. These isostructural frameworks have undergone extensive study in which diverse properties and phenomena have been explored such as Jahn-Teller and spin-crossover phase transitions[23,24], magnetic properties[19,25], and electrical conductivity[26,27]. It has been revealed that the $[Fe(ta)_2]$ framework, when exposed to inert gases like Ar or $CO_2$, exhibits conductivity changes, primarily due to alterations in the MOF's deformation potential caused by guest loading at specific sites[28,29]. Furthermore, the modular nature of $[M(ta)_2]$ allows for accurate pore size adjustments by substitution of metal ions with different radii, including $Mn^{II}$ and $Zn^{II}$, resulting in the size control of features within 0.01 nm and influencing the interaction with gas molecules. Therefore, the tuneable pore size and quadrupole moments within $[M(ta)_2]$ make it particularly suitable for $H_2/D_2$ separation[22].

Extensive research has been conducted on the adsorption of various gases, including Ar, $CO_2$, and $N_2$, in these frameworks[30,31]. These studies have yielded insights into the mechanisms of adsorption, successfully pinpointing different adsorption sites and their occupancy rates, and capacity. However, despite extensive studies, identifying $H_2$ adsorption sites within this particular MOF remains challenging. Computational potential energy surface (PES) analysis suggests that hydrogen molecules may localize both inside and outside the framework's pockets[22]. Despite extensive characterization of gas adsorption behavior in triazolate-based frameworks, a critical knowledge gap exists regarding the molecular-level understanding of hydrogen isotope interactions and the framework's structural response to these interactions.

Herein, we investigated the penetration of hydrogen molecules into these pore-surface pockets by employing neutron powder diffraction (NPD) using deuterated $[Mn(ta)_2]$. Our in situ diffraction experiments during $H_2$ and $D_2$ dosing at various temperatures revealed two adsorption sites in the void space of the material: one at the side pockets and the other within the channels. These experiments demonstrated that gas loading induces expansion of the unit cell volume, with a more pronounced effect observed for $H_2$ compared to $D_2$ for the same adsorbed amount. We further investigated the direct separation of hydrogen isotopes by cryogenic thermal desorption spectroscopy (TDS). In this study, we have gained detailed molecular-level information on $H_2$ and $D_2$ binding in $[Mn(ta)_2]$ and studied the structural dynamics around adsorption sites and their properties, allowing us to reconcile these details with TDS and NPD results. These characterization efforts provide clear insights into the molecular aspects of $H_2/D_2$ adsorption and selectivity in this Mn-triazole MOF, which will help in designing materials with the desired properties for efficient isotope separation.

## Results and discussion
### Materials and characterization
High-quality crystals of $[Mn(ta)_2]$ and $[Mn(ta-d_2)_2]$ were synthesized according to literature procedures, using 1H-1,2,3-triazole (H-ta) and $(4,5-D_2)-1H-1,2,3$-triazole (H-ta-$d_2$) ligands in a 3:1 ligand-to-metal ratio[21]. The deuterated version of $[Mn(ta)_2]$ was used for NPD experiments to suppress the incoherent neutron scattering arising from the H in the framework that obscures the coherent neutron scattering important to obtaining structural information, allowing for accurate determination of hydrogen binding sites. X-ray powder diffraction

(XRPD) data of both compounds align well with existing literature (see Supplementary Figs. 1, 2) indicating the material is phase-pure, and scanning electron microscopy (SEM) images reveal the typical octahedral shape of the $M(ta)_2$-type framework crystals (Supplementary Figs. 3, 4). Fourier transform infrared spectroscopy (FT-IR) shows significant differences between the spectrum of $[Mn(ta)_2]$ and $[Mn(ta-d_2)_2]$, attributable to the heavier deuterium isotope (Supplementary Figs. 5–7). These differences align with those in the spectrum of $[Zn(ta)_2]$ and $[Zn(ta-d_2)_2]$ simulated using the code CASTEP, as similar to previous work and confirming successful synthesis of the material with the deuterated ligand[32,33]. Weak additional bands in the $[Mn(ta-d_2)_2]$ spectrum associated with the non-deuterated and mixed-isotope ligand are in fair agreement with the purity of the commercially available H-ta-$d_2$ ligand. Successful activation of $[Mn(ta-d_2)_2]$ was confirmed by thermogravimetric analysis (TGA), see Supplementary Fig. 8, which involved removing solvent molecules from the synthesis to prepare the material for gas adsorption. This analysis revealed no residual solvent, demonstrating the robustness of the material's framework to desolvation.

### Gas adsorption properties
Isothermal gas adsorption by $Mn(ta)_2$ exhibited a two-step process for both $N_2$ and Ar (Supplementary Figs. 9, 10). For $N_2$, saturation was observed at 125 and 246 $cm^3$/g (STP, standard temperature, and pressure), corresponding to 1.0 and 1.9 $N_2$ molecules per Mn, respectively. Similarly, for Ar, saturation occurred at 125 and 265 $cm^3$/g (STP), corresponding to 1.0 and 2.1 Ar molecules per Mn, respectively. These results indicate a consistent two-step uptake process for both gases, aligning with values previously reported. In contrast, $H_2$ and $D_2$ adsorption at various temperature (30 – 77 K) exhibited a single-step process, i.e., a type I adsorption curve that is typical for nanoporous materials. These type I $H_2$ isotherms at 77 K have been previously observed in metal-triazolates[22]. Saturation was observed at 15.8 and 16.0 mmol/g for $H_2$ and $D_2$, respectively, corresponding to ~3 $H_2$ molecules per Mn and 3 $D_2$ molecules per Mn at 30 K (Fig. 1a, b, Supplementary Fig. 11). At all temperatures, $D_2$ uptake was slightly higher than $H_2$, indicating preferential adsorption of $D_2$. Interestingly, the isotherms showed a similar uptake at 1 bar when the temperature increased from 30 to 60 K, decreasing to only 2 $H_2(D_2)$/Mn at 77 K. This behavior of $[Mn(ta)_2]$ is uncommon, as most other materials typically show a significant decrease in uptake at 1 bar with increasing temperature. To understand this unusual behavior, the isosteric heat of adsorption was calculated using the Clausius-Clapeyron equation applied to the adsorption isotherms. The obtained adsorption heat ranged from 4.8 to 5.8 kJ/mol for $H_2$, while for $D_2$, a near-constant heat of adsorption of -6.5 kJ/mol was observed, increasing with surface coverage, which refers to the fraction of the surface occupied by adsorbate molecules (Fig. 1c, d). The gradually increasing adsorption energy suggests that the uptake of $H_2/D_2$ becomes more favorable at higher loadings, which is contrary to the usual trend. The nearly constant heat of adsorption across the entire gas loading range indicates possible structural changes upon adsorption, exposing more favorable binding sites.

### In situ high-resolution neutron powder diffraction
To elucidate this unusual adsorption behavior, comprehensive in situ high-resolution NPD experiments were performed using the Echidna instrument at the Australian Centre for Neutron Scattering (ACNS) at the Australian Nuclear Science and Technology Organisation (ANSTO). These experiments provided a better understanding of the material's site-specific interaction with $H_2$ and $D_2$. Rietveld refinement using the diffraction data for the activated material confirms that $[M(ta)_2]$ $[Mn(ta-d_2)_2]$ crystallizes with cubic symmetry with a $Fd\bar{3}m$ space group (No. 227) (Supplementary Tables 1, 2). N donor atoms octahedrally coordinate each Mn from surrounding 1,2,3-triazole to form a

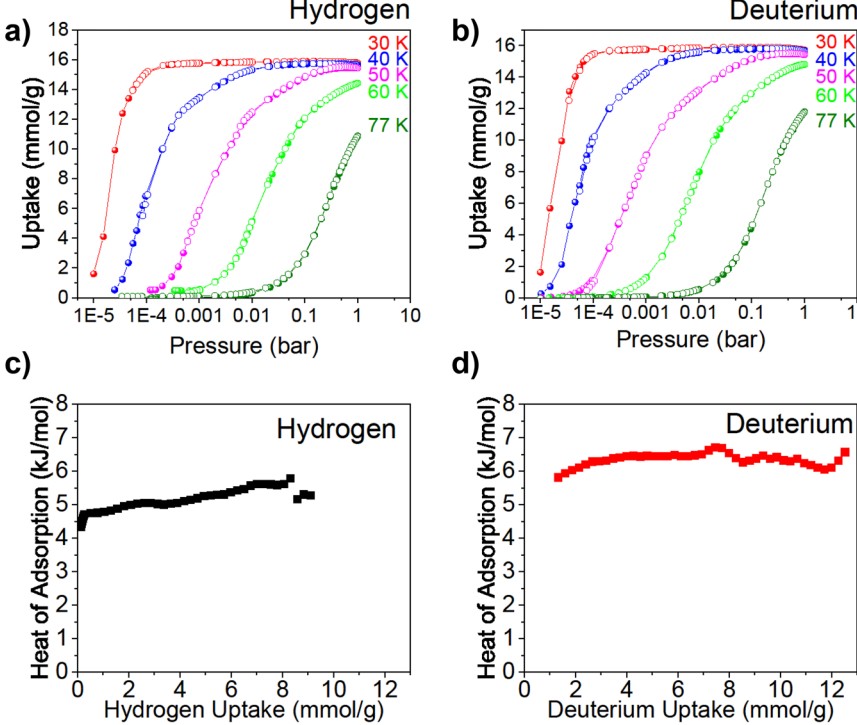

**Fig. 1 | H$_2$ and D$_2$ adsorption properties. a** Hydrogen and (**b**) deuterium in [Mn(ta)$_2$] at various temperatures (30–77 K) in the pressure range of 0–1 bar (adsorption and desorption, full and open points, respectively). Lines through the points are a guide to the eye. The isosteric heat of adsorption of (**c**) hydrogen and (**d**) deuterium in [Mn(ta)$_2$] as a function of the hydrogen/deuterium gas uptake. Source data are provided as a Source Data file.

diamond-type framework. Mn(1) occupies the 8$b$ Wyckoff site (symmetry $\bar{4}3m$) and is coordinated only by N donor atoms of the $\mu_3$-bridging triazolate linker. Mn(2) is situated at the 16$d$ Wyckoff site (symmetry $\bar{3}m$), bonded exclusively to N1 or N3 donor atoms (Supplementary Figs. 12, 13). The pore network of the determined structure runs along the [110] direction, aligning well with that reported from XRPD data of non-deuterated samples[21,22].

bvAfter Rietveld refinement of the host structure, the sample was dosed with 15 mmol/g of D$_2$ and H$_2$ gas at 40 K and further NPD data were collected at 15 K. Rietveld refinement of the structure in conjunction with the examination of the residual nuclear density arising from the long-range ordering of adsorbed H$_2$ or D$_2$ using Fourier-difference methods enabled the determination of the position of adsorbed H$_2$ or D$_2$ in the framework structure. This revealed two adsorption sites, as depicted in Fig. 2 and Supplementary Figs. 14–16. Site 1 is located inside the pockets formed by the ta-ligands, with the closest framework atoms being the D atoms of the ligand. Site 2 is situated in the pore channel along the [110] direction. Each site, with a multiplicity of 192, can accommodate a maximum of 32 D$_2$/H$_2$ molecules per unit cell, leading to a structure with an orientational disorder of the linkers when saturated with D$_2$ or H$_2$ (Supplementary Fig. 17). The occupation of D$_2$ or H$_2$ at both Site 1 and Site 2 in the saturated material is 1.4 H$_2$ (D$_2$) per Mn, consistent with the 3 H$_2$(D$_2$) per Mn at saturation reached in the adsorption isotherms. The closest distance between adsorbed H$_2$ and the framework at Site 2 is ~3.48 Å, while that for H$_2$ at Site 1 is ~2.48 Å (the distance between the nearest D atom of the ta ligand and the center of mass of the adsorbed H$_2$) (Supplementary Table 3). Relative to Site 1, the larger distance between the host and H$_2$ at Site 2 relative to Site 1 is typical of a weaker interaction. Moreover, the distance between hydrogen molecules adsorbed at Site 1 and 2 lies in the range 3.63 – 3.84 Å. In the case of D$_2$, the adsorption sites are identical to those for H$_2$.

In addition to the two adsorption sites for H$_2$ and D$_2$ identified, we note an expansion of the [Mn(ta-d$_2$)$_2$] crystal volume, with the material saturated with H$_2$ and D$_2$ being ~1.2% and 1%, respectively, larger than for the empty material. Specifically, the lattice parameter expands from 18.08 to 18.14 Å upon adsorption of D$_2$, which is slightly less than the expansion caused by H$_2$, increasing from 18.08 to 18.15 Å. This smaller lattice expansion for D$_2$ suggests more favorable adsorption of the heavier guest molecules, likely due to the stronger interactions between the D$_2$ molecules and the adsorbent.

The structural dynamics observed in the Mn(ta-d$_2$)$_2$ framework upon hydrogen loading are in good agreement with the structure predicted using density functional theory (DFT). The introduction of H$_2$ molecules into the framework induces a moderate expansion of both lattice parameters and unit-cell volume, detailed data is provided in the Supplementary Section Density Functional Calculations. The refined structure of the empty framework is detailed in Supplementary Table 4 and Supplementary Fig. 19 depicts the progressive increase in volume associated with hydrogen loading. The empty unit-cell volume expands by ~1.3% upon uptake of 18.0 mmol/g H$_2$, in good agreement with the isotherm and NPD data. Though this value is slightly higher than the experimental uptake, it likely reflects the assumption of a perfect-crystal model in the theoretical framework. This expansion primarily originates from an elongation of ~0.65% in the manganese-nitrogen bond lengths, while the bond lengths within the triazole rings remain unchanged, as shown in Supplementary Fig. 20.

**Time-resolved neutron powder diffraction**

To further elucidate structural dynamics in the framework as a function of H$_2$ and D$_2$ loading and temperature, time-resolved NPD data were collected using the high-intensity neutron diffractometer Wombat at the ACNS. Data were collected every minute consecutively during isothermal gas adsorption and desorption. Datasets for H$_2$ and D$_2$ adsorption and desorption were recorded at 30 and 60 K and

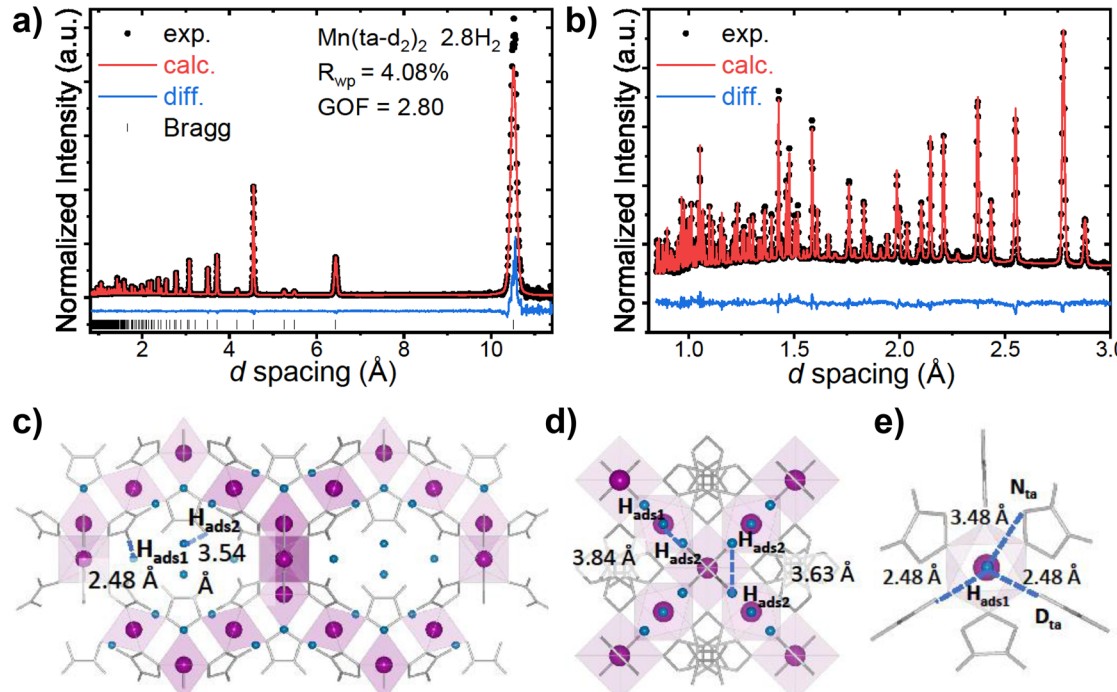

**Fig. 2 | High-resolution neutron powder diffraction. a** Rietveld refinement profile using NPD data, collected at 15 K for [Mn(ta-$d_2$)$_2$], loaded with 15 mmol/g hydrogen at 40 K, where contributions from the vanadium sample holder were excluded from the refinement. Figures of merit shown include the goodness of fit (GOF) and the weighted profile R-factor ($R_{wp}$). **a** Entire data range; **b** Low-d section of data. **c** Schematic of the refined [Mn(ta)$_2$] structure showing molecular hydrogen at the two adsorption sites in the framework, viewed along the [110]. Mn atoms are shown as magenta spheres, the Mn-N polyhedra as light magenta, the ta-ligand as gray five-membered aromatic rings with H excluded for clarity, and adsorbed H$_2$ molecules as blue spheres where Site 1 is denoted H$_{ads1}$ and Site 2 is denoted H$_{ads2}$, (**d**) view along the [100], (**e**) refined short-range average structure of H$_{ads1}$ viewed along the [111]. Source data are provided as a Source Data file.

confirmed the two adsorption sites previously identified using NPD data collected under equilibrium conditions. The rapid collection of diffraction data during adsorption and desorption processes revealed distinct adsorptive behaviors for H$_2$ and D$_2$ at the specified sites, as shown in Fig. 3. At a loading temperature of 30 K, as shown in Fig. 3a, b, H$_2$ molecules were observed to simultaneously occupy both adsorption sites following a 2 mmol/g loading. Similar adsorption behavior was noted for D$_2$, with occupancy at both Site 1 and Site 2. The unit cell volume of [Mn(ta)$_2$] at 30 K is plotted as a function of H$_2$ and D$_2$ loading in Fig. 3c. The data reveal an expansion in the unit cell with increasing gas loading, with the framework unit cell volume expanding by ~1.3% upon adsorption of 13 mmol/g of H$_2$, with a slightly lesser expansion for D$_2$ (1.1%).

At 60 K, [Mn(ta)$_2$] exhibits distinct adsorption mechanisms for H$_2$ and D$_2$. For H$_2$, a sequential pore-filling process is observed: initially, H$_2$ molecules exclusively occupy Site 1 (smaller pockets) at low loadings. Occupation of Site 2 (channels) commences only after reaching 8 mmol/g (~1.5 H$_2$/Mn), with adsorption continuing until near-saturation at ~3 H$_2$ molecules per Mn. In contrast, D$_2$ adsorption occurs simultaneously at both Site 1 and Site 2, with Site 2 reaching 50% occupancy when Site 1 is fully saturated at 8 mmol D$_2$/g (Fig. 3d, e).

The unit cell volume evolution of [Mn(ta)$_2$] at 60 K (Fig. 3f) displays a similar profile to that observed at 30 K, despite the different adsorption mechanisms. The framework undergoes a significant expansion for both isotopes, with the unit cell volume increasing by ~1.35% upon adsorption of 11 mmol/g of H$_2$. D$_2$ induces a slightly smaller expansion of 1.2% at an equivalent loading. Analysis of the adsorption data reveals that this expansion is predominantly driven by H$_2$/D$_2$ occupation at Site 1, with minimal volume changes occurring during the occupation of isotope gases at Site 2. This site-specific expansion behavior elucidates the slightly larger overall expansion

observed at 60 K compared to 30 K. The differential impact of Site 1 and Site 2 occupation on framework expansion provides crucial insights into the structural dynamics underlying the adsorption process and may contribute to the material's isotope separation capabilities. However, the exact correlation between site occupation and lattice expansion remains to be fully elucidated. As evidenced in Fig. 2, the adsorption sites within the framework exhibit distinct geometries and short-range environments, likely contributing to the complex interplay between gas loading, site occupation, and structural deformation.

These findings highlight the intricate nature of gas-framework interactions in [Mn(ta)$_2$] and their potential implications for hydrogen isotope separation. The subtle difference in adsorption behavior of H$_2$ and D$_2$, coupled with the framework's structural response, may be the key to understand the material's high isotope separation efficiency. Further studies employing advanced in situ characterization techniques and theoretical modeling could provide deeper insights into the molecular-level mechanisms underlying these observations and their impact on separation efficiency.

## Isotope separation performance

Direct separation of isotopic H$_2$/D$_2$ mixtures was measured using a custom-built cryogenic thermal desorption spectroscopy (TDS) system. We exposed the sample to 100 mbar of 1:1 mixture of H$_2$:D$_2$ for 10 min at various temperatures ($T_{exp}$) ranging from 30 to 60 K and TDS data was collected between 20 and 110 K (Fig. 4 and Supplementary Fig. 21). In TDS data, the desorption curve area is directly proportional to the number of adsorbed gas molecules, allowing quantification of the amount of adsorbed gas with the D$_2$/H$_2$ selectivity ($S_{D2/H2}$) derived from the peak area ratio. Figure 4c displays the D$_2$/H$_2$ selectivity and corresponding D$_2$ uptake as a function of $T_{exp}$, revealing a decreasing

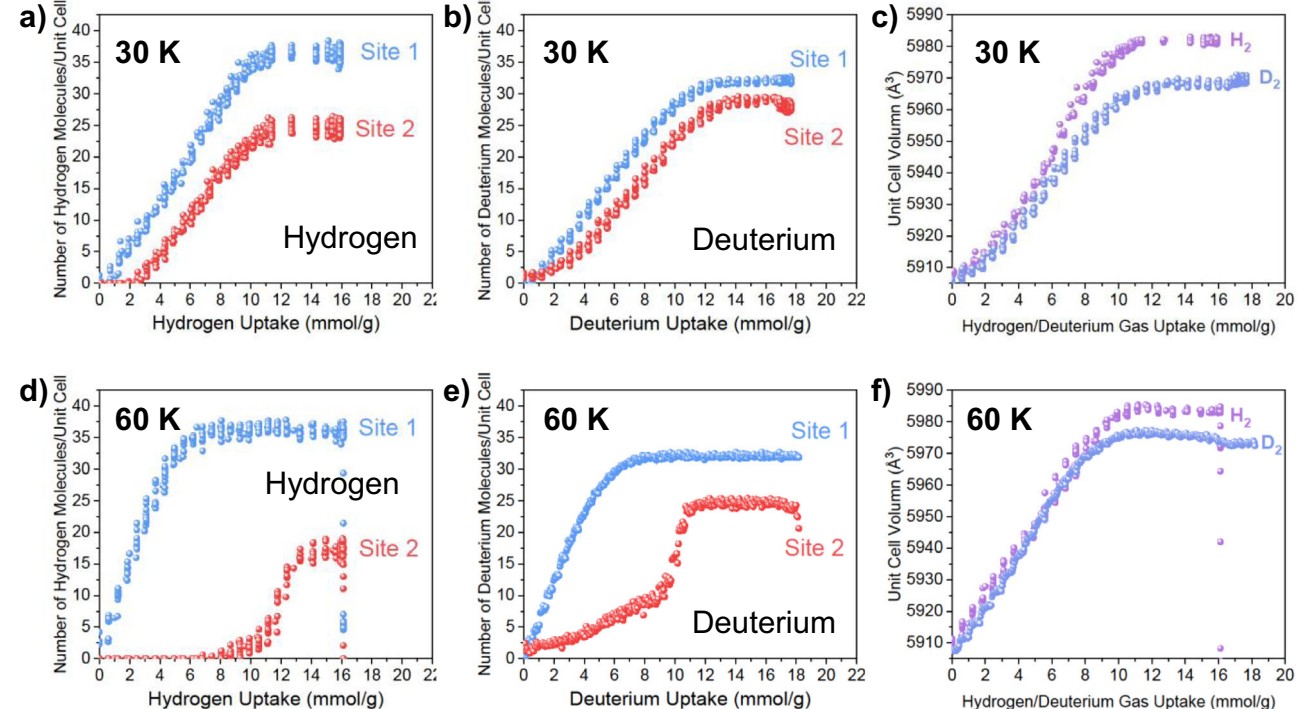

**Fig. 3 | Gas induced site occupancy and framework expansion. (a–f)** Data collected at 1-min intervals during isothermal gas adsorption and desorption. Top row (**a–c**): Results at 30 K; bottom row (**d–f**): Results at 60 K. Left column (**a**, **d**): Occupation of $H_2$ at Site 1 (blue, pockets) and Site 2 (red, channels) as a function of total $H_2$ uptake. Middle column (**b**, **e**): Occupation of $D_2$ at Site 1 (blue, pockets) and Site 2 (red, channels) as a function of total $D_2$ uptake. Right column (**c**, **f**): Unit cell volume expansion of [Mn(ta)$_2$] as a function of total gas uptake for $H_2$ (purple) and $D_2$ (light blue). Error bars are smaller than the symbol size. Source data are provided as a Source Data file.

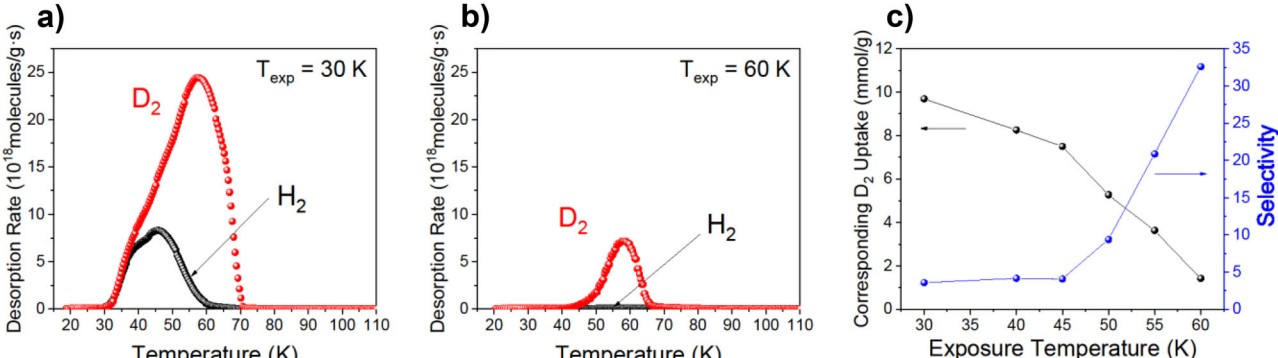

**Fig. 4 | Isotope separation TDS measurements.** $H_2$ (black) and $D_2$ (red) thermal desorption data during exposure of [Mn(ta)$_2$] to 100 mbar of a 1:1 mixture of $H_2$:$D_2$ for 10 min at various temperatures ($T_{exp}$): (**a**) 30 K and (**b**) 60 K. **c** The corresponding amount of adsorbed $D_2$ (black) and selectivity (blue) as function of $T_{exp}$. Lines through the points are a guide to the eye. Source data are provided as a Source Data file.

uptake but increasing selectivity with temperature, peaking at 60 K ($S_{D2/H2} = 32.5$). This value represents one of the highest selectivities reported for porous materials under similar conditions, as confirmed by a comparison with previously reported $D_2$/$H_2$ separation materials (Supplementary Table 5). We also examined the effect of gas pressure on isotope separation at 60 K using a 1:1 mixture of $H_2$:$D_2$ pressures from 10 to 300 mbar (Supplementary Fig. 22), observing consistent preferential adsorption of $D_2$. Exposure of the sample to a 5:95 mixture of $D_2$:$H_2$, reflecting the natural abundance of $D_2$, at 100 mbar at 60 K for 10 min resulted in a selectivity of 74 (Fig. 5), with $D_2$ concentration reaching nearly 75% in just one cycle, highlighting Mn-triazolate's potential as a highly effective material for industrial deuterium separation.

## Structural dynamics and separation mechanism

To understand the exceptional separation performance of our framework, we conducted additional in situ NPD experiments, using deuterated [Mn(ta)$_2$], at POWGEN, a high-resolution NPD instrument located at the Spallation Neutron Source (SNS) at Oak Ridge National Laboratory (ORNL), focusing on isotope exchange at 40 K. To validate the reliability of the NPD data, we first compared the isotope separation performance of the deuterated sample with that of the non-deuterated sample under identical conditions. The TDS data, illustrated in Supplementary Fig. 23, demonstrate comparable separation performance for both the deuterated and non-deuterated ligand samples. Initially, the framework was loaded with 15 mmol/g $H_2$ at 40 K, showing an average occupancy of 1.4 $H_2$ molecules per adsorption site.

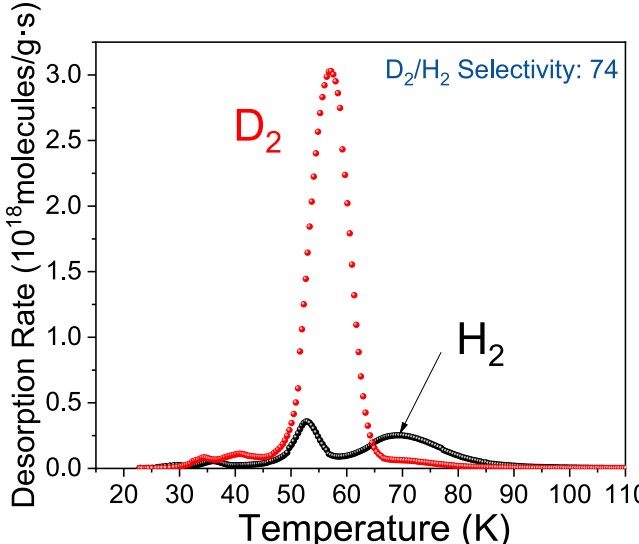

**Fig. 5 | Separation of diluted isotope mixture.** TDS spectra of $H_2$ (black)/$D_2$ (red) after exposure of [Mn(ta)$_2$] to a 100 mbar mixture of 95 : 5 $H_2$ : $D_2$ for 10 min at 60 K. Source data are provided as a Source Data file.

We introduced $D_2$ to the framework at the same temperature to observe the exchange behavior between $H_2$ and $D_2$ within the adsorption sites. NPD data were collected continuously during this process allowing for real-time monitoring of the changes in the scattering length density of atoms residing at the two adsorption sites. Hydrogen and deuterium atoms have scattering lengths of opposite sign, and therefore having deuterium substituting hydrogen at the same site would lead to a reduction in the refined nominal hydrogen occupancy. The refinement revealed a significant reduction in the H occupancy, suggesting an increase in scattering length, at both Site 1 and Site 2. Particularly, Site 1 exhibited a more pronounced decrease in $H_2$ occupancy compared to Site 2. This indicates that $H_2$ molecules at Site 1 are more susceptible to replacement by $D_2$ molecules, suggesting a higher propensity for isotope exchange at this site.

This change is also clearly observed in the residual nuclear density map, where a gradual reduction in negative nuclear density indicates the replacement of $H_2$ by $D_2$ (data shown at 30, 60, and 90 min, Supplementary Figs. 24, 25), providing a clear representation of the isotope exchange process. This exchange is significant because strong adsorption sites are typically associated with open metal sites in other materials[13]. Unlike many other frameworks that rely on undersaturated metal sites for strong adsorption, our framework features fully-coordinated metal sites. Our findings reveal that the confinement within the structure itself, rather than the presence of open metal sites, acts as a potent framework feature for gas adsorption.

In summary, our results reveal a complete and detailed experimental study of both the structural and hydrogen isotope adsorption behavior of the triazolate-based MOF [Mn(ta)$_2$], possessing pores with a diameter of ~6.1 Å. Hydrogen and deuterium adsorption by the material is characterized by a single-step type I isotherm with a saturation uptake of 3 $H_2$($D_2$)/Mn, while $N_2$ and Ar exhibit a lower uptake and a two-step isotherm. The isosteric heat of adsorption for $H_2$ and $D_2$ remains relatively constant during increasing adsorption. High-resolution and high-intensity neutron powder diffraction identified two distinct adsorption sites for $H_2$ and $D_2$, with Site 1 located at the entrance of the pocket formed by triazole ligands and Site 2 located in the pore channels. Gas loading induces expansion of the unit cell which can be attributed to Mn-N bond elongation in line with theoretical predictions. Time-resolved neutron powder diffraction during in situ gas dosing revealed occupation of both sites by $D_2$ and $H_2$

simultaneously at 30 K. At the higher loading temperature of 60 K, population of Site 1 and 2 by $D_2$ occurs more quickly, with a faster population at Site 1 and further population of Site 2 occurring after saturation of Site 1. In contrast, at 60 K, $H_2$ occupation at Site 2 begins only after Site 1 is fully occupied. Based on this subtle isotopic difference in structural dynamics, our work highlights the potential for developing novel porous frameworks with high selectivity for hydrogen isotopes. Thermal desorption spectroscopy experiments following exposure of the sample to $H_2$/$D_2$ mixtures at 60 K confirm the very high $D_2$/$H_2$ selectivity of 32.5 and separation from a 95:5 mixture of $H_2$:$D_2$ with a 75% recovery of $D_2$ in one cycle. The observed high $D_2$/$H_2$ selectivity coincides with differences in the isotopologue-induced structural dynamics of the framework upon adsorption of these isotopes. While our data indicate a correlation between the slightly lower volume expansion for $D_2$ and its preferential adsorption, attributing the high selectivity solely to the structural dynamics would be an oversimplification. Multiple phenomena, including quantum sieving effects, differences in zero-point energy, and subtle variations in gas-framework interactions (gas-driven unit cell expansion), likely contribute to the observed selectivity. Further studies, such as detailed computational modeling of isotope-framework interactions or isotope-specific vibrational spectroscopy, could provide more direct evidence of the mechanisms underlying this high selectivity. Understanding the interplay between framework confinement, gas-solid interactions, and structural responses remains an important area for future research in isotope separation using metal-organic frameworks. The commercial availability of the ligand and ease of synthesizing dia-topology MOFs with various divalent metals make industrial-scale production and application of these materials for isotope separation feasible.

## Methods

### Materials synthesis

N,N-Diethylformamide (DEF; 99%; TCI), N,N-dimethylformamide (DMF; 99.8% analytical grade; VWR), methanol (MeOH; 99.8% analytical grade; VWR), 1H-1,2,3-triazole (1,2,3-triazole; 98%; BLD Pharmatech Ltd.), (4,5-$D_2$)−1H-1,2,3-triazole (1,2,3-triazole-4,5-$d_2$, 95%, CombiPhos Catalysts, Inc.), and manganese(II) nitrate tetrahydrate (Mn(NO$_3$)$_2$·4H$_2$O; 98.5%; Merck) were used as received from the commercial suppliers.

[**Mn(ta)$_2$**] Mn(NO$_3$)$_2$·4H$_2$O (3.20 g, 12.75 mmol), dissolved in 160 mL of DEF, and 1H-1,2,3-triazole (2.40 mL, 2.86 g, 41.41 mmol) were added into a 400 ml inner volume ACE round-bottom pressure flask and capped with a silicon O-ring at the front seal of the polytetrafluoroethylene (PTFE) bushing. The mixture was heated to 120 °C in an oven for 3 days. The product was filtered and washed successively with DMF (3 × 20 mL) and MeOH (3 × 20 mL). Drying overnight under vacuum at room temperature (RT) afforded the phase pure product as a white powder (950 mg, 39%).

[**Mn(ta-d$_2$)$_2$**] Mn(NO$_3$)$_2$·4H$_2$O (1.17 g, 4.66 mmol), dissolved in 57 mL of DEF, and 1H-1,2,3-triazole-4,5-$d_2$ (1.00 g, 14.07 mmol) were added into a 200 ml inner volume ACE pressure tube and capped with a silicon O-ring at the front seal of the PTFE bushing. The mixture was heated to 120 °C in an oven for 3 days. The product was filtered and washed successively with DMF (3 × 10 mL) and MeOH (3 × 10 mL). Drying overnight under vacuum at RT afforded the phase pure product as a white powder (0.86 g, 63%).

### Material characterizations

**Thermogravimetric analysis (TGA).** TGA of a 6.207 mg sample of the as-synthesized [Mn(ta-d$_2$)$_2$] was prepared at ambient conditions and measured at the University of Augsburg with a TA Instruments Q500 device in a temperature range of 25–800 °C under a nitrogen atmosphere with a heating rate of 5 °C min$^{-1}$ starting after a 5 min isothermal step.

**Fourier-transform infrared spectroscopy (FTIR)**. FTIR spectra were measured at the University of Augsburg under ambient conditions on a Bruker Equinox 55 FT-IR spectrometer equipped with a PLATINUM ATR unit and a KBr beam splitter in the wavenumber range 4000 – 400 cm$^{-1}$. Spectra in the range 1800 – 180 cm$^{-1}$ were recorded on the same instrument equipped with a Si beam splitter. The signals were labeled strong (s), medium (m), weak (w), and very weak (vw).

**X-ray powder diffraction (XRPD)**. XRPD data of the as-synthesized samples were collected at the University of Augsburg under ambient conditions with an Empyrean (PANalytical) diffractometer equipped with a Bragg–Brentano$^{HD}$ mirror, PIXcel$^{3D}$ 2 × 2 detector using Cu Kα$_1$ radiation. The Mn(ta)$_2$ and Mn(ta-d$_2$)$_2$ samples were measured with a step size of 0.0016° in the 2θ range 3°–60° and 3°–120° and measurement times of 37882 s (10.52 h) and 58445 s (16.23 h), respectively.

**Gas sorption isotherm measurements**

A fully automated Sieverts apparatus iQ2 (Quantachrome Instruments) was used to perform gas adsorption experiments. High-purity hydrogen (H$_2$, 99.999%) and deuterium (D$_2$, 99.8%) gases were used throughout the experiments. Corrections to the sample volume and the non-linearity of the adsorbate were made using data for an empty analysis carried out at the same temperature and pressure range as for samples. Approximately 30 mg of sample was heated to 393 K under vacuum overnight to remove solvent molecules. A cryocooler based on the Gifford-McMahon cycle was used to control the sample temperature from 20–300 K with an estimated error of <0.05 K.

The isosteric heat of adsorption of H$_2$ and D$_2$ in the samples was independently determined by isothermal gas adsorption and thermal desorption spectroscopy (TDS). H$_2$ and D$_2$ adsorption isotherms were measured in the low-pressure region from 30 K to 77 K. The heat of adsorption was determined by the isosteric method. Data were transferred into Van't Hoff plots for different coverages, where the isosteric heat of adsorption is defined as the negative of the isosteric enthalpy of adsorption. For a given surface coverage, the isosteric enthalpy of adsorption can be derived from the so-called Van't Hoff equation, analogous to the Clausius-Clapeyron equation. The pressure of equilibrium (P$_{eq}$) for a fixed amount of adsorbed gas (n$_a$) was determined by linear interpolation of the data points of the adsorption isotherms at different temperatures. A linear regression of the slope for the relation ln(P$_{eq}$) versus 1/T was used to calculate the isosteric heat of adsorption.

**Thermal desorption spectroscopy (TDS) studies of hydrogen isotope separation**

The selective adsorption after exposure to D$_2$/H$_2$ isotope mixtures was measured by an in-house designed thermal desorption spectroscopy (TDS) system[34,35]. For a typical process, ~3.5 mg sample was loaded in the sample holder and activated at 393 K under vacuum for 5 h. An equimolar D$_2$/H$_2$ isotope (purity H$_2$ 99.999% and D$_2$ 99.8%) mixture was exposed to the activated sample at a fixed temperature T$_{exp}$ for an exposure time t$_{exp}$. The free gas was evacuated and the sample cooled to 20 K to preserve the adsorbed state. The sample was then heated to room temperature at 0.1 K/s during which the desorbing gas was continuously detected using a mass spectrometer (QMS), noting a pressure increase in the sample chamber as the gas desorbed. The area under the desorption peak is proportional to the desorbed gas amount, and calibration of the TDS apparatus enabled its determination.

Calibration of the mass spectrometer was carried out using ~0.5 g solid Pd$_{95}$Ce$_5$ etched with aqua regia to remove any oxide layer. The alloy was heated to 600 K under a high vacuum to remove any hydrogen absorbed during the etching procedure, after which it was exposed to 40 mbar pure H$_2$ or pure D$_2$ over 1.5 – 2.5 h at 350 K, with the mass recorded. As H and D are bound preferentially to Ce atoms at low exposure pressure, the alloy was handled under ambient conditions for a short time. The alloy was weighed after cooling to room temperature, and the mass difference between the unloaded and loaded state being equal to the mass uptake of hydrogen or deuterium, respectively. After weighing, the alloy was loaded in the chamber again and heated at 0.1 K · s$^{-1}$ from room temperature to 600 K and another desorption spectrum measured in which the area under the desorption peak directly corresponds to the mass uptake of gas.

**In situ neutron powder diffraction**

In situ neutron powder diffraction (NPD) experiments were conducted using the Echidna and Wombat diffractometers at the Australian Centre for Neutron Scattering (ACNS) at the Australian Nuclear Science and Technology Organisation (ANSTO). Echidna utilized a Ge(335) monochromator at a takeoff angle of 140°, yielding a wavelength of 1.62207(1) Å, with no pre-sample collimation and fixed tertiary 5′ collimation. Wombat used a Ge(115) monochromator at a takeoff angle of 90°, resulting in a wavelength of 1.54298(1) Å, with no pre-sample collimation and tertiary oscillating collimation.

The exact wavelength and instrumental contributions to the data were obtained using data for the National Institute of Standards and Technology (NIST) standard reference material (SRM) 660b (La$^{11}$B$_6$). Data analysis was carried out using the program TOPAS7, and for both instruments, the peak profile was characterized by a convolution of an axial divergence model with one refinable asymmetry parameter, a constant hat function with a refinable breadth parameter, and a TCHZ-pseudo Voigt function in which the refinable parameters are U, V, W, and Y.

Data were collected for a known mass of polycrystalline desolvated [Mn(ta-d$_2$)$_2$], which was pre-treated at 1 × 10$^{-4}$ mbar at 110 °C and transferred to a sealed 9 mm diameter vanadium can inside a helium-filled glove box. The sample can with temperature sensors at the top and bottom was attached to a custom-designed gas delivery sample stick, which was positioned within a top-loading cryo-furnace, keeping the sample isolated from air. A Hiden Isochema IMI manometric dosing system was used to control gas delivery/vacuum to the sample, and the top-loading cryostat was used to control the sample temperature. This experimental setup is reported elsewhere[36–38].

At Echidna, NPD data of [Mn(ta-d$_2$)$_2$] were collected at -15 K for the evacuated framework and the samples dosed with 2, 5, 7.5, and 15 mmol/g of D$_2$ or H$_2$ gas at all temperatures. During all collections, the top of the sample was -19 K and the bottom was 15 K. Each collection had a slightly different time length, but most were ~7 h.

At Wombat, NPD data of Mn(ta)$_2$ were collected every minute during adsorption at 30 K, 55 K, and 60 K of D$_2$ and H$_2$. H$_2$ and D$_2$ adsorption isotherms were performed at 30, 55, and 60 K, each consisting of 19 pressure steps in the range 0.2 to 700 mbar. Minimum and maximum wait times for equilibration at each pressure point were 3 and 10 min, respectively. NPD data were collected continuously during the isotherm experiments and recorded at 1 min intervals. Timestamps for the NPD data frames were converted to the corresponding guest uptake amount by linear interpolation of the time-dependent uptake data recorded by the manometric system at irregular 1–20 s intervals.

All NPD data were analyzed using the Rietveld method as implemented in TOPAS7. A starting model for the crystal structure of Mn(ta)$_2$ was obtained from an isostructural compound Fe(II)(ta)$_2$ reported by Grzywa et al.[23]. For all high-resolution data at 15 K, 6 profile parameters (as used to describe data for the La11B6 standard reference material 660b), the cubic lattice parameter, the zero shift, 6 positional parameters, 5 isotropic atomic displacement parameters for all atoms in the asymmetric unit, and 10 Chebyshev background polynomial parameters were refined in the angular range from 6 to 155° 2θ. In the case of the crystal structure of Mn(ta)$_2$ without any adsorbed gas, the refinement converged quickly. To model the disorder in the H$_2$/D$_2$

loaded samples, the ta-d linker group was constrained to be a rigid unit, allowing only rotational degrees of freedom. Complete refinement parameters, atomic coordinates, and thermal factors are provided in Supplementary Information NPD analysis, Tables 1, 2.

Additional NPD data were collected using the POWGEN diffractometer at the Spallation Neutron Source (SNS), Oak Ridge National Laboratory (ORNL). Approximately 0.6 g of sample was loaded into a 6 mm diameter cylindrical vanadium sample can. Data were collected for ~3 h in high-resolution mode using a center wavelength of 1.5 Å, covering a d-spacing range from 0.50 Å to 12.0 Å. The experiment was carried out using a top-loading JANIS cryofurnace. The unloaded pristine sample was first measured at 6 K to obtain the structure of the framework without adsorbed gas. Gas loading was performed using an in-house built manifold at 40 K, with either pure $H_2$ or $D_2$. After loading, the sample was cooled to 6 K and diffraction data was collected to determine the structure of the gas-loaded framework.

The isotopic exchange measurements were conducted using a sequential gas-loading protocol. First, hydrogen was introduced to the sample at 40 K and allowed to equilibrate for 30 min, establishing a baseline of $H_2$ occupancy throughout the framework. The system was then cooled to 6 K, where initial NPD data were collected to accurately determine the $H_2$ adsorption sites and occupancies. Subsequently, the sample was warmed back to 40 K, and an equivalent amount of $D_2$ gas was introduced to the system. To monitor the progressive isotope exchange process, NPD data were collected at three time intervals (30, 60, and 90 min) after $D_2$ introduction, allowing direct visualization of the replacement of $H_2$ by $D_2$ at specific adsorption sites.

Rietveld analysis of the NPD data was performed using the GSAS-II software package[39]. The peak profile was modeled using a convolution of a Gaussian peak shape function and a back-to-back exponential function to account for peak asymmetry. The instrumental profile parameters were determined by refining the data from a standard silicon sample (NIST SRM 640 d)[40]. The background was modeled using a shifted Chebyshev polynomial function with 8 terms.

### Density Functional Theory (DFT) calculations

All calculations were performed using density functional theory (DFT) with the Perdew-Burke-Ernzerhof (PBE) functional[41], employing Goedecker-Teter-Hutter (GTH) pseudopotentials[42] with scalar-relativistic core corrections. Long-range dispersion interactions were accounted for using the D3 London dispersion correction scheme[43]. A triple-zeta valence plus polarization (TZVPP) basis set was used, with a grid cutoff of 400 Ry for geometry optimization, as implemented in the QUICKSTEP module[44] of the CP2K program package[45]. The framework structures were modeled using periodic boundary conditions in all three dimensions.

The empty framework structure was first optimized with respect to atomic positions and lattice parameters. This optimized structure served as the starting configuration for all loaded structures. Hydrogen molecules ($H_2$) were introduced into the accessible volume of the framework, with initial positions and orientations randomly assigned using the Packmol code. Subsequent geometry optimization was performed, allowing both atomic positions and lattice parameters to relax. The resulting total energy of the loaded structure was recorded. A single-point calculation was then carried out on the emptied structure (after $H_2$ removal) to determine its total energy. This procedure is schematically illustrated in Supplementary Fig. 18 and has been previously described[15].

To account for potential variations in adsorption sites and molecular orientations, statistical averaging was conducted over 25 independent configurations, each with randomly assigned initial positions and orientations of the $H_2$ molecules. The deviations in adsorption energies across these configurations were found to be negligibly small.

## Data availability
The data underlying graphs generated in this study are provided in the Source Data file. All data supporting the findings of this study are available within the article and the Supplementary Information. The raw experimental data generated during the study are available from the corresponding authors upon request. Source data are provided with this paper.

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

## Acknowledgements

L.Z. is grateful for partial funding by the Japan Society for the Promotion of Science (JSPS) Grants-in-Aid for Scientific Research (KAKENHI) (No. JP24K17650). We gratefully acknowledge the Australian Nuclear Science and Technology Organisation (ANSTO; proposal number P9673) and the Spallation Neutron Source (SNS) at Oak Ridge National Laboratory (ORNL; proposal number IPTS-29081.1) for providing the neutron beamtime essential to this research. J.-O. J. thanks the Zentrum für Informationsdienste und Hochleistungsrechnen (ZIH) at TU Dresden for granted computational time within project TRANSPHEMAT. L. Z. thanks Prof. Hao Li from Tohoku University, for his suggestions, proof-reading, and constructive feedback.

## Author contributions

R.R.-O. synthesized and characterized the materials and contributed to writing the manuscript. L.Z. performed the H$_2$/D$_2$ sorption and separation measurements, data acquisition, and analysis, and led the preparation and revising of the manuscript. V.K.P. and S.D. performed neutron diffraction experiments at ANSTO and V.K.P. contributed to revising the manuscript. R.D. performed the analysis of the neutron diffraction data obtained from ANSTO and contributed to the writing of the manuscript. L.Z. and C.L. performed neutron diffraction experiments at ORNL and C.L. contributed to the analysis of the neutron diffraction data. J.L.F. and J.-O. J. performed the DFT computational studies and participated in writing the manuscript. D.V. supervised the material synthesis and general characterization and contributed to the revision of the manuscript. M.H. led the preparation and revision of the manuscript, and supervised and coordinated this study.

## Funding

## Competing interests

The authors declare no competing interests.
