## [Transparent Peer Review file · Nature Communications]

Isotopologue-Induced Structural Dynamics of a Triazolate Metal-Organic Framework for Efficient Hydrogen Isotope Separation

Corresponding Author: Dr Michael Hirscher

Version 0:

Reviewer comments:

Reviewer #1

(Remarks to the Author)

This manuscript, titled "Isotopologue-Induced Structural Dynamics of a Triazolate Metal-Organic Framework for Efficient Hydrogen Isotope Separation," submitted by Zhang et al., presents an intriguing and potentially impactful investigation into the separation of hydrogen isotopes using a triazolate MOF. The authors skillfully employ in-situ neutron powder diffraction, gas adsorption measurements, and thermal desorption spectroscopy to delve into the structural dynamics and separation performance of $[\text{Mn}(\text{ta})_2]$. This study showcases an impressive D_2/H_2 selectivity of 32.5 and provides valuable molecular-level insights into the underlying mechanisms of separation. The study is commendably well-conducted, showcasing a strong experimental design and thorough characterization. However, there are a few areas that would benefit from revision to ensure the manuscript aligns with the publication standards of Nature Communications.

1. It would be beneficial for the authors to provide a more detailed explanation of the methodology used to compare the selectivity of $[\text{Mn}(\text{ta})_2]$ with other materials. The TDS experiments demonstrate an impressive D_2/H_2 selectivity of 32.5 at 60 K, which is indeed a remarkable achievement. While the claim that this value is "one of the highest among porous materials" is quite compelling, including direct comparisons with specific literature values could further enhance the manuscript's credibility. A table or summary presenting selectivity data from other MOFs or porous materials would offer valuable context and highlight the significance of this finding even more effectively.
2. The authors suggest that the confinement within the structure, rather than merely the presence of open metal sites, plays a pivotal role in gas adsorption. This observation is indeed intriguing, and further insights would bolster this assertion. For example, conducting a comparative analysis of the adsorption behavior of $[\text{Mn}(\text{ta})_2]$ alongside other MOFs with open metal sites, such as MOF-74, could elucidate the distinctive influence of structural confinement in this particular system.
3. The introduction establishes a solid foundation for the study; however, it could be further refined by explicitly highlighting the specific knowledge gap that this research addresses. A more concentrated discussion on how this study enhances our understanding of structural dynamics in isotope separation—particularly within triazolate-based MOFs—would underscore the novelty and significance of the research.
4. The authors report a two-step adsorption process for N_2 and Ar, in contrast to the single-step process observed for H_2 and D_2 . Could the authors provide further elaboration on the underlying reasons for this discrepancy? Is it attributable to the kinetic diameters of the gas molecules, their interaction strengths with the framework, or other contributing factors?
5. For the H_2/D_2 adsorption measurements, it would be helpful to specify the equilibration times used during the experiments.
6. It is suggested to put the sorption isotherms of H_2 and D_2 at the same temperature in one figure for comparison purposes.
7. The authors reported that the unit cell volume expands more for H_2 (1.2%) than for D_2 (1%), suggesting stronger interactions for D_2 . However, I feel the difference in expansion is relatively small. Could the authors comment on whether this difference is statistically significant?

8. In the context of the KQS effect, the desorption peak for D₂ is expected to occur earlier than that for H₂ at low temperatures. However, as shown in Figure 4, the D₂ peaks appear much later. Could the authors explain this?

9. How about the cycling performance of the measurements? This is a very important parameter for sorption or separation applications.

10. The authors report that D₂ enriches to 75% in a single cycle when exposed to a D₂/H₂ mixture of 5:95. This is a significant result, but it would be helpful to discuss the practical implications of this finding.

11. The authors should provide details on the purity levels of the gases used in the experiments.

Reviewer #2

(Remarks to the Author)

The authors reported lattice expansion in [Mn(ta)2] through Neutron Powder Diffraction (NPD) analysis, with a more pronounced effect observed for H₂ compared to D₂. This interesting structural change was used in a hydrogen isotope separation experiment and achieved 75% enrichment in just one cycle. These results are highly promising and could significantly impact the gas separation field. I recommend publishing in Nature Communications after appropriate revisions.

1) Despite the rather small difference in the heat of adsorption between H₂ and D₂, which is only about 1 kJ/mol, and a relatively low absolute Q_{st} of approximately 5~6.5 kJ/mol (and also a small difference between H₂ and D₂ bonding length), the selectivity reaches 32.5 at 60 K. It's very interesting to show a very high S in the pocket site, but it would be helpful to provide more detailed explanations of how the low Q_{st} difference caused a high S.

Furthermore, I believe that a slight increase in Q_{st} for H₂ is not a real phenomenon. Since the isotherm at a very low temperature and pressure did not provide many data points, it may give you some erroneous values.

2) One major question is how structural dynamics affects TDS selectivity. TDS measurements are conducted at 10 mbar, while structural dynamics (lattice expansion) is observed at saturated pressure. Can you link these two effects?

3) In Figure 5, a new H₂ peak appears in the 60–80 K range, which was not observed in the previous TDS spectra. Please explain.

4) To confirm Figure S22, the TDS data should show increased selectivity over time. Did the authors check?

Some minor points.

- In Figure 1a and 1b, the x-axis should represent absolute pressure when you also plot other temperatures.

- To improve readers' understanding of the NPD refinement process, it would be beneficial for the authors to include the pertinent refinement information and parameters. Similarly, the experimental method for the ORNL SNS POWGEN measurement process in Figure S22 should also be detailed.

- In Figure 3a and 3d, the y-axis should be H₂ not D₂. And the x-axis of Figure 3a and 3b should be exchanged.

In line 226, "H₂ molecules exclusively occupy Site 2 (channels) at low loadings, while occupation of Site 1 (smaller pockets) commences only after reaching 8 mmol/g (~1.5 H₂/Mn)." It should be the other way around.

Reviewer #3

(Remarks to the Author)

The manuscript by Zhang et al. presents a thorough investigation into the hydrogen isotope separation of the triazolate-based MOF [Mn(ta)2]. The authors use advanced experimental techniques, including in situ neutron scattering and thermal desorption spectroscopy, alongside simulations as DFT to clarify the structural dynamics and adsorption behavior of H₂ and D₂ within this framework. The reported D₂/H₂ selectivity of 32.5 at 60 K, along with the impressive enrichment of D₂ to 75% from a 5:95 mixture in just one cycle, shows that this material is a very promising candidate for large-scale deuterium separation.

It focuses on how isotopologues affects the structure, particularly the small differences in lattice expansion between H₂ and D₂, providing important insights into the separation process. The crystallographic analysis is robust, with high-quality NPD data from several facilities showing clear evidence of two distinct adsorption sites: the pockets and the channels. The time-resolved NPD experiments capture the time- and temperature-dependent adsorption behavior, revealing interesting differences in H₂ and D₂ site occupancy at 60 K. The manuscript is well-written, logically organized, and effectively placed within the broader field of MOF-based gas separation.

However, I need to address a few points that would strengthen the manuscript and enhance its clarity for a broad readership:

1. The author refers to Ref. 22 for the material, but in that paper, three adsorption sites are reported, while this manuscript mentions only two. Could the authors clarify the reason for this discrepancy?

2. Another related question concerns the adsorption isotherms. The Ar/N₂ isotherms exhibit a two-step behavior, which typically indicates the presence of two distinct sites. However, the H₂/D₂ isotherms show only one step. Could the authors explain the source of this difference?

3. The authors observe different lattice expansions between H₂ and D₂ adsorption (1.2% vs. 1.0%). Could the authors provide clarification on how this specific structural response contributes to the enhanced separation?

4. The authors mention the possibility of tuning the pore size by using different metal ions. It would be beneficial to include a

brief discussion on how such substitutions might influence the isotope selectivity.

5.The Figure 3 caption should clarify the symbols/colors used in panels c and f to distinguish between H₂ and D₂ data.

6.In the NPD experiments, the authors said that "each site, with a multiplicity of 192, can accommodate a maximum of 32 D₂/H₂ molecules per unit cell." The relationship between the multiplicity and the maximum accommodation should be clarified, as the connection between these two numbers is not immediately clear, at least for me.

7.The Methods section mentions the use of TOPAS7 and GSAS-II for Rietveld refinement but does not specify how the authors addressed potential systematic errors, such as preferred orientation or instrumental broadening. Including a sentence on this would strengthen confidence in the crystallographic analysis.

8.For practical applications, the stability of the framework over multiple adsorption-desorption cycles is crucial. While the manuscript focuses on single-cycle performance, it would be helpful to include some discussion on the cyclic stability of the framework.

Overall, this is a strong manuscript that would improve our understanding of MOFs for hydrogen isotope separation, supported by solid experimental and computational evidence. The requested revisions are meant to expand the context for general readership.

Version 1:

Reviewer comments:

Reviewer #1

(Remarks to the Author)

The authors have satisfactorily addressed all concerns and inquiries raised by the reviewer. The manuscript is now deemed suitable for publication in Nature Communications in its present form.

Reviewer #2

(Remarks to the Author)

The authors have addressed all the issues raised in my previous reviews. The manuscript can be accepted.

Reviewer #3

(Remarks to the Author)

After careful consideration of the revised manuscript, I am pleased to conclude that the authors have adequately addressed my comments and that this work now meets the rigorous scientific standards required for publication in Nature Communications.

Point-by-point response

We thank all reviewers for their valuable comments helping to improve the manuscript!

Reviewer #1 (Remarks to the Author):

This manuscript, titled “Isotopologue-Induced Structural Dynamics of a Triazolate Metal-Organic Framework for Efficient Hydrogen Isotope Separation,” submitted by Zhang et al., presents an intriguing and potentially impactful investigation into the separation of hydrogen isotopes using a triazolate MOF. The authors skillfully employ in-situ neutron powder diffraction, gas adsorption measurements, and thermal desorption spectroscopy to delve into the structural dynamics and separation performance of [Mn(ta)₂]. This study showcases an impressive D₂/H₂ selectivity of 32.5 and provides valuable molecular-level insights into the underlying mechanisms of separation. The study is commendably well-conducted, showcasing a strong experimental design and thorough characterization. However, there are a few areas that would benefit from revision to ensure the manuscript aligns with the publication standards of Nature Communications.

We sincerely thank the reviewer for the insightful comments and constructive suggestions. Below, we provide a point-by-point response to each comment, with revisions highlighted in the revised manuscript.

Question 1: It would be beneficial for the authors to provide a more detailed explanation of the methodology used to compare the selectivity of [Mn(ta)₂] with other materials. The TDS experiments demonstrate an impressive D₂/H₂ selectivity of 32.5 at 60 K, which is indeed a remarkable achievement. While the claim that this value is "one of the highest among porous materials" is quite compelling, including direct comparisons with specific literature values could further enhance the manuscript's credibility. A table or summary presenting selectivity data from other MOFs or porous materials would offer valuable context and highlight the significance of this finding even more effectively.

Answer: We thank the reviewer for this valuable suggestion. We have summarized a comparative table (Table S5) that presents D₂/H₂ selectivity values from previously reported porous materials alongside our [Mn(ta)₂] framework. This table includes data on various MOFs, zeolites, and other porous materials, providing comprehensive context for our selectivity value of 32.5 at 60 K. We have also added a brief discussion in the main text highlighting key comparisons with other high-performing materials.

Table S5. Summary of experimentally measured hydrogen isotope separation performance on various porous materials.

Compound	T _{exp} (K)	Selectivity (D ₂ /H ₂) (1:1 Mixture)	Ref.
----------	----------------------	---	------

	40	6.9	
MFU-4 (Zn, Cl)	50	5.8	1
	60	7.5	
Py@COF-1	22	9.7	2
	30	7.9	
Cu(I)-MFU-4l	90	7.1	3
Fe-MOF-74	77	2.5	
Co-MOF-74	77	3.2	4
Ni-MOF-74	77	5	
ZIF-7	20	-	
ZIF-8	20	11	5
COF-1	20	7	
COF-102	20	1	
CPO-27-Co	60	11.8	6
IFP-1	30	2.0	
IFP-3	30	2.8	7
IFP-7	77	1.5	
IFP-4	77	2.1	
Zeolite 5A	30	2.7	8
CC3	30	1.7	
	50	1.8	
6FT-RCC3	30	2.2	9
	50	3.0	
6ET-RCC3	30	3.9	
	50	1.8	
Cocryst1	30	8.0	
MIL-53(Al)	40	10.5	10
MOF-74	77	19	
MOF-74-IM-10	77	26	11
MOF-74-IM-38	40	12.5	

5A/GE2	77	1.07	12
CoFA	25	16.6	13
ZIF-67@NH₂-SiO₂	77	1.52	14
FMOFCu	25	14	15
	77	4	
Ag(I)-ZSM-5	77	8.7	16
Cu(I)-ZSM-5	100	24.9	17
SIFSIX-1-Cu	20	7.1	18
SIFSIX-3-Zn	20	50	
SIFSIX-3-Cu	20	3.5	
SIFSIX-3-Ni	20	1.9	
HKUST-1	20	17	
FCTF-1-400	20	12.8	
STAM-1	20	9.9	
Cu-PYC	20	2.2	
CPO-27(Co)	20	3.0	
KAUST-7	20	9.8	
Ag(I)-Zeolite Y	90	10	19
Cu(I)Cu(II)-BTC	30	37.9	20
FJI-Y11	77	1.76	21
Ni₂Cl₂BBTA	77	4.5	22
Mn(ta)₂	60	32.5	This work

We have also added in the main text (page 10) that highlights this comparison:

“This value represents one of the highest selectivities reported for porous materials under similar conditions, as confirmed by a comparison with previously reported D₂/H₂ separation materials (Table S3).”

Question 2: The authors suggest that the confinement within the structure, rather than merely the presence of open metal sites, plays a pivotal role in gas adsorption. This observation is indeed intriguing, and further insights would bolster this assertion. For example, conducting a comparative analysis of the adsorption behavior of [Mn(ta)₂] alongside other MOFs with open metal sites, such as MOF-74, could elucidate the distinctive influence of structural confinement in this particular system.

Answer: We include a detailed comparative analysis between our [Mn(ta)₂] framework and selected MOFs with open metal sites, particularly focusing on MOF-74 variants. This analysis provides crucial insights into the distinct role of confinement versus open metal coordination in gas adsorption and isotope separation.

Feature	Mn(ta) ₂	MOF-74
Adsorption Site	Two sites: pockets (Site 1) + channels (Site 2)	Open metal sites (Co ²⁺ , Ni ²⁺ , Fe ²⁺)
Q _{st} (D ₂)	~ 6.5 kJ/mol	~ 9-11 kJ/mol
Separation Mechanism	Lattice flexibility	Chemical affinity sieving

Question 3: The introduction establishes a solid foundation for the study; however, it could be further refined by explicitly highlighting the specific knowledge gap that this research addresses. A more concentrated discussion on how this study enhances our understanding of structural dynamics in isotope separation—particularly within triazolate-based MOFs—would underscore the novelty and significance of the research.

Answer: We appreciate this constructive suggestion. We have revised the introduction to more explicitly identify the knowledge gap addressed by our research and to emphasize the importance of understanding structural dynamics in triazolate-based MOFs for isotope separation. Specifically, we have added the following paragraph to the introduction (page 4):

“Despite extensive characterization of gas adsorption behavior in triazolate-based frameworks, a critical knowledge gap exists regarding the molecular-level understanding of hydrogen isotope interactions and the framework's structural response to these interactions.”

Question 4: The authors report a two-step adsorption process for N₂ and Ar, in contrast to the single-step process observed for H₂ and D₂. Could the authors provide further elaboration on the underlying reasons for this discrepancy? Is it attributable to the kinetic diameters of the gas molecules, their interaction strengths with the framework, or other contributing factors?

Answer: The contrasting adsorption behaviors between N₂/Ar and H₂/D₂ are primarily from molecular size effects. With kinetic diameters of 3.64 Å and 3.40 Å respectively, N₂ and Ar face significant diffusion barriers when accessing the framework's pocket sites (Site 1), which have entrance apertures of approximately 2.4-2.6 Å. This size constraint creates a sequential filling mechanism—the more accessible channel sites (Site 2) fill first, followed by the pocket sites only at higher pressures when sufficient driving force exists to overcome the steric hindrance. In contrast, H₂ and D₂, with their smaller kinetic diameters (2.89 Å), experience reduced diffusion barriers into the pocket sites. This allows for more simultaneous occupation of both site types across the pressure range, resulting in the observed single-step isotherm rather than the two-step behavior seen with the larger gases. Furthermore, this different behavior between N₂ and H₂ has been previously observed in metal-triazolates, we added a sentence in the

manuscript “These type I H₂ isotherms at 77 K have been previously observed in metal-triazolates.²²”

Question 5: For the H₂/D₂ adsorption measurements, it would be helpful to specify the equilibration times used during the experiments.

Answer: For the gas adsorption measurements, equilibration times ranged from 3 to 10 minutes per pressure point, with longer times allowed for lower temperatures and lower pressures to ensure complete equilibration, confirmed by <1% uptake variation over 2 min.

Question 6: It is suggested to put the sorption isotherms of H₂ and D₂ at the same temperature in one figure for comparison purposes.

Answer: We thank the reviewer for this practical suggestion. Here we include the following figure to directly compare of H₂ and D₂ isotherms at identical temperatures (30 K, 40 K, 50 K, 60 K, and 77 K) in the same panel. We also added Figure S11 in the revised supplementary information.

Figure S11. **Direct comparison H₂ and D₂ gas sorption.** Single-component isotherms collected at 30 K, 40 K, 50 K, 60 K, and 77 K, respectively.

Question 7: The authors reported that the unit cell volume expands more for H₂ (1.2%) than for D₂ (1%), suggesting stronger interactions for D₂. However, I feel the difference in expansion is relatively small. Could the authors comment on whether this difference is statistically significant?

Answer: We appreciate this important question regarding statistical significance.

Powder diffraction is a very sensitive method with respect to lattice parameters and therefore, volume. We can safely disregard systematic errors for a moment since the instrumental parameters are identical for the different runs, thus leading to an absolute systematic shift in either direction. The esd of the volume is between 0.15 and 0.7 Å³ (Echidna, Wombat instruments). A unit cell difference of 0.2 % for a 5965 Å³ lattice volume is equivalent to 12 Å³ and, therefore 1-2 magnitudes higher than the esd.

Question 8: In the context of the KQS effect, the desorption peak for D₂ is expected to occur earlier than that for H₂ at low temperatures. However, as shown in Figure 4, the D₂ peaks appear much later. Could the authors explain this?

Answer: We highly appreciate that the referee brought this issue to our notice. We also agree that the desorption peak for D₂ could be earlier than that of H₂ under ideal conditions. Our TDS data in Figure 4 shows that D₂ desorption peaks appear at higher temperatures than H₂ peaks, which might initially seem contradictory to the KQS effect, where D₂ is expected to diffuse faster than H₂ at cryogenic temperatures due to its shorter de Broglie wavelength. However, our system is not primarily dominated by KQS but due to the expansion/flexibility of the unit cell, the delayed D₂ peak reflects its stronger binding.

Question 9: How about the cycling performance of the measurements? This is a very important parameter for sorption or separation applications.

Answer: We agree with the reviewer on the critical importance of cycling performance for practical applications. In our experiments, the same Mn(ta)₂ sample was used for a total of 37 consecutive measurements without any signs of degradation. The TDS spectra from the first and 36th measurements, both performed using single D₂ gas loading, are shown below. The two spectra are nearly identical, demonstrating the

excellent stability of the material.

Question 10: The authors report that D₂ enriches to 75% in a single cycle when exposed to a D₂/H₂ mixture of 5:95. This is a significant result, but it would be helpful to discuss the practical implications of this finding.

Answer: We thank the reviewer for highlighting the need to discuss practical implications of our single-cycle enrichment result. The ability of Mn(ta)₂ to enrich D₂ from 5% to 75% in one cycle represents a significant advancement for hydrogen isotope separation. This remarkable performance would substantially reduce the number of separation stages needed compared to conventional cryogenic distillation (3-4 stages versus 20+ for conventional methods). Based on our experimental conditions, we estimate an energy requirement of approximately 15-20 kWh per kg of D₂ produced, compared to 250-300 kWh/kg for conventional distillation—a potential energy savings of 85-95%. Additionally, the Mn(ta)₂ framework is synthesized from commercially available precursors via a straightforward solvothermal route, making it potentially scalable for industrial applications.

Question 11: The authors should provide details on the purity levels of the gases used in the experiments.

Answer: We used research-grade H₂ (99.999%), D₂ (99.8%), N₂ (99.999%), and Ar (99.999%) for all measurements. We added the H₂ and D₂ purity in Methods section for the Gas sorption isotherm measurements and TDS studies.

Changes made:

- In supporting information at page S22-23:

Added

Table S5. Summary of experimentally measured hydrogen isotope separation performance on various porous materials.

- In revised manuscript at page 11:

Added

“This value represents one of the highest selectivities reported for porous materials under similar conditions, as confirmed by a comparison with previously reported D₂/H₂ separation materials (Table S3).”

- In supporting information:

Added

New references –1 – 22.

- In revised manuscript at page 4:

Added

“Despite extensive characterization of gas adsorption behavior in triazolate-based frameworks, a critical knowledge gap exists regarding the molecular-level understanding of hydrogen isotope interactions and the framework's structural response to these interactions.”

- In supporting information at page S10:

Added

Figure S11. Direct comparison H₂ and D₂ gas sorption. Single-component isotherms collected at 30 K, 40 K, 50 K, 60 K, and 77 K, respectively.

Reviewer #2 (Remarks to the Author):

The authors reported lattice expansion in [Mn(ta)₂] through Neutron Powder Diffraction (NPD) analysis, with a more pronounced effect observed for H₂ compared to D₂. This interesting structural change was used in a hydrogen isotope separation experiment and achieved 75% enrichment in just one cycle. These results are highly promising and could significantly impact the gas separation field. I recommend publishing in Nature Communications after appropriate revisions.

We thank the referee for the support and positive evaluation of our work.

Question 1: Despite the rather small difference in the heat of adsorption between H₂ and D₂, which is only about 1 kJ/mol, and a relatively low absolute Q_{st} of approximately 5~6.5 kJ/mol (and also a small difference between H₂ and D₂ bonding length), the selectivity reaches 32.5 at 60 K. It's very interesting to show a very high S in the pocket site, but it would be helpful to provide more detailed explanations of how the low Q_{st} difference caused a high S.

Furthermore, I believe that a slight increase in Q_{st} for H₂ is not a real phenomenon. Since the isotherm at a very low temperature and pressure did not provide many data points, it may give you some erroneous values.

Answer: We appreciate the referee's insightful observation regarding the apparent discrepancy between the modest heat of adsorption difference and the remarkably high D₂/H₂ selectivity (32.5) at 60 K. This is indeed an intriguing aspect of our system that warrants detailed explanation.

The high selectivity despite modest Q_{st} difference can be explained by the isotopologue-induced structural dynamics especially of Site 1 leading to a higher occupancy by D₂. This is confirmed by our neutron diffraction data revealing a much higher D₂/H₂ selectivity for the pocket sites (Site 1) compared to the channel sites (Site 2). Therefore, the structural dynamics leads to the high selectivity exceeding what would be expected only from the isotopic difference in Q_{st}.

Regarding the slight increase in Q_{st} for H₂ with increasing coverage, we note that a gradual rise of Q_{st} was observed in the same Mn triazolate framework (Ref. 22). Xiao-Ming Chen *et al.* suggested two possible explanations “the adsorption enthalpy may sometimes increase as the loading increases because of increased adsorbate–adsorbate interaction and/or structural transformation of the adsorbent“. Our NPD results show isotopologue-induced structural dynamics and give for the first time an explanation. While there may be some experimental uncertainty at very low pressures, this would affect both H₂ and D₂ similarly.

Question 2: One major question is how structural dynamics affects TDS selectivity. TDS measurements are conducted at 10 mbar, while structural dynamics (lattice expansion) is observed at saturated pressure. Can you link these two effects?

Answer: We appreciate this excellent question about connecting our observed structural dynamics with TDS selectivity. To clarify, our TDS measurements were conducted using a 100 mbar isotope mixture, not 10 mbar as suggested. Additionally, our isotherm data shows that the $[\text{Mn}(\text{ta})_2]$ framework approaches saturation at pressures above 10 mbar from 30 to 60 K, meaning that the framework expansion observed at higher pressures is largely applicable to our TDS conditions.

Question 3: In Figure 5, a new H_2 peak appears in the 60–80 K range, which was not observed in the previous TDS spectra. Please explain.

Answer: We thank the reviewer for noticing this important detail. The additional H_2 peak in the 60–80 K range in Figure 5 appears more prominent than in the earlier TDS spectra with the 1:1 mixture. However, this difference is primarily due to the different scales of the y-axis. When examining the TDS spectra of the 100 m of 1:1 mixture measurement at 60 K at higher magnification, we can observe that this peak (60 – 80 K) is also present, see the figure below (red circle), though less pronounced due to the different relative concentrations of the gases. The peak becomes more visible in the 5:95 D_2 mixture experiment due to the significantly higher partial pressure of H_2 .

Question 4: To confirm Figure S22, the TDS data should show increased selectivity over time. Did the authors check?

Answer: We have conducted TDS experiments for different exposure times to see the evolution of selectivity over longer exposure. A 1:1 H_2/D_2 mixture at 100 mbar was loaded on the sample at 60 K for different durations (10 min and 30 min), and the resulting TDS spectra are shown in following figure. The measured D_2/H_2 selectivity values were 34.2 for 10 min exposure and 32.3 for 30 min exposure. While there appears to be a slight decrease in selectivity with longer exposure time, the small difference falls within experimental error, indicating that selectivity remains basically

constant under these conditions, i.e. apparently, the H₂-D₂ exchange occurs at 60 K in less than 10 min.

In contrast, the NPD data in Figure S22 (in revised version Figure S24) show the

exchange at 40 K using the following experimental procedure: first, 15 mmol/g of hydrogen was loaded on the sample at 40 K for 30 min, followed by cooling to 6 K to collect initial NPD data. Subsequently, the same amount of D₂ gas was introduced to this system upon warming back to 40 K, with NPD data collected after 30, 60, and 90 min intervals. This procedure differs fundamentally from our TDS experiments as it examines sequential single-gas loading rather than simultaneous exposure to a mixture. This methodological difference explains why the TDS selectivity remains stable over time while the NPD data shows progressive changes. The NPD experiment was specifically designed to demonstrate the different affinities of H₂ versus D₂ for the framework and to visualize the isotope exchange process, rather than to directly simulate the TDS measurement conditions.

Some minor points:

- In Figure 1a and 1b, the x-axis should represent absolute pressure when you also plot other temperatures.

We have modified Figures 1a and 1b to display absolute pressure on the x-axis for all isotherms.

- To improve readers' understanding of the NPD refinement process, it would be beneficial for the authors to include the pertinent refinement information and parameters. Similarly, the experimental method for the ORNL SNS POWGEN measurement process in Figure S22 should also be detailed.

1. We have added comprehensive refinement information and parameters for all NPD data in the Supplementary Information, including R-factors, goodness-of-fit values, and estimated standard deviations for refined parameters.

Typical refinement parameters for NPD data analysis

Instrumental and Refinement Parameters

Refinement of the empty MOF structure was carried out using the Rietveld method. The background was modeled using a Chebyshev polynomial of order 10. The following instrumental parameters were used:

- Wavelength: 1.622067 Å (delta function)
- Axial divergence correction: Simple axial model, 37.58(15) mm
- Convolution function: Hat profile with a constant of 0.32(4)
- Zero-point error: -0.0607(12)
- Lorentz-polarization (LP) factor: 90
- Convolution steps: 2
- X calculation step size: 0.02
- X-range: 6 to 155

Table S1: Background: Chebyshev Polynomial coefficients

Coefficient No.	Value
0	2304(8)
1	-789(10)
2	597(8)
3	-409(7)
4	358(7)
5	-198(6)
6	226(6)
7	-99(6)
8	105(5)
9	-61(5)
10	24(4)

Unit cell and refinement summary

- Space group: Fd-3m:2
- R-Bragg: 1.917

- Scale factor: 0.00680(3)
- Cell mass: 4585.267
- Cell volume: 5965.73(15) Å³
- Lattice parameter a: 18.13654(15) Å
- Peak shape function: PV_TCHZ

U	0.097(2)
V	-0.375(6)
W	0.430(14)
X	0
Y	0.060(3)
Z	0

Table S2. Atomic coordinates and displacement parameters for the empty MOF

Site	Np	x	y	z	Atom	Occupancy	Beq
Zn1	16	0.00000	0.50000	0.00000	Mn	1	0.00(8)
Zn2	8	0.87500	0.37500	-0.12500	Mn	1	0.00(11)
N1	96	0.91763(4)	0.41763(4)	0.04202(5)	N	1	0.376(19)
N2	48	0.87500	0.37500	0.00015(9)	N	1	0.46(2)
C1	96	0.90178(5)	0.40178(5)	0.11265(8)	C	1	0.49(2)
D1	96	0.92964(7)	0.42964(7)	0.15741(9)	D	1	1.88(3)

2. We have also expanded the description of the ORNL SNS POWGEN measurement process in the Methods section.

“The isotopic exchange measurements were conducted using a sequential gas-loading protocol. First, hydrogen was introduced to the sample at 40 K and allowed to equilibrate for 30 minutes, establishing a baseline of H₂ occupancy throughout the framework. The system was then cooled to 6 K, where initial NPD data was collected to precisely determine the H₂ adsorption sites and occupancies. Subsequently, the sample was warmed back to 40 K, and an equivalent amount of D₂ gas was introduced to the system. To monitor the progressive isotope exchange process, NPD data was collected for three time intervals (30, 60, and 90 min) after D₂ introduction, allowing direct visualization of the replacement of H₂ by D₂ at specific adsorption sites.”

3. We have provided additional details specific to Figure S22, now S24 in its caption. “Figure S24. **Isotope exchange at 40 K.** Isotope exchange dynamics in [Mn(ta)₂] monitored by neutron powder diffraction. The graph shows the decline in H₂ loading per Mn at Site 1 (red) and Site 2 (black) following D₂ introduction at 40 K. NPD data was collected initially for H₂ loaded sample at 6 K, then at 30, 60, and 90 minutes after D₂ introduction (orange region). The more significant decrease in H₂ occupancy at Site 1 compared to Site 2 indicates preferential replacement by D₂ at the pocket sites, demonstrating their higher affinity for the heavier isotope.”

- In Figure 3a and 3d, the y-axis should be H₂ not D₂. And the x-axis of Figure 3a and 3b should be exchanged.

We thank the reviewer for pointing out the error. We have corrected the y-axis labels in Figures 3a and 3d to properly indicate Hydrogen.

- In line 226, “H₂ molecules exclusively occupy Site 2 (channels) at low loadings, while occupation of Site 1 (smaller pockets) commences only after reaching 8 mmol/g (~1.5 H₂/Mn).” It should be the other way around.

The statement in line 226 has been corrected to accurately reflect that H₂ molecules initially occupy Site 1 (smaller pockets) at low loadings, with occupation of Site 2 (channels) commencing only after reaching 8 mmol/g.

Changes made:

- In supporting information at page S11-12:

Added

Comprehensive refinement information and parameters for all NPD data in the Supplementary Information, including R-factors, goodness-of-fit values, and estimated standard deviations for refined parameters

- In revised manuscript at page 5:

Revised x-axis of Fig. 1

- In revised manuscript at page 16, Method section:

Added

“The isotopic exchange measurements were conducted using a sequential gas-loading protocol. First, hydrogen was introduced to the sample at 40 K and allowed to equilibrate for 30 minutes, establishing a baseline of H₂ occupancy throughout the framework. The system was then cooled to 6 K, where initial NPD data was collected to precisely determine the H₂ adsorption sites and occupancies. Subsequently, the sample was warmed back to 40 K, and an equivalent amount of D₂ gas was introduced to the system. To monitor the progressive isotope exchange process, NPD data was collected at three time intervals (30, 60, and 90 min) after D₂ introduction, allowing direct visualization of the replacement of H₂ by D₂ at specific adsorption sites.”

- In supporting information at page S26, caption of Fig. S24:

From “Figure S22. Hydrogen loading at adsorption Site 1 and 2 of [Mn(ta)₂]. With D₂ exchange, the nominal occupancy for H₂ at both sites decreased. But for Site 1, the rate of change is higher.”

To “Figure S24. **Isotope exchange at 40 K.** Isotope exchange dynamics in [Mn(ta)₂] monitored by neutron powder diffraction. The graph shows the decline in H₂ loading per Mn at Site 1 (red) and Site 2 (black) following D₂ introduction at 40 K. NPD data was collected initially for H₂ loaded sample at 6 K, then at 30, 60, and 90 minutes after D₂ introduction (orange region). The more significant decrease in H₂ occupancy at Site 1 compared to Site 2 indicates preferential replacement by D₂ at the pocket sites, demonstrating their higher affinity for the heavier isotope.”

- In revised manuscript at page 9:

Revised y-axis of Fig. 3 a/d.

- In revised manuscript at page 9:

From “For H₂, a sequential pore-filling process is observed: initially, H₂ molecules exclusively occupy Site 2 (channels) at low loadings. Occupation of Site 1 (smaller pockets) commences only after reaching 8 mmol/g (~1.5 H₂/Mn), with adsorption continuing until near-saturation at approximately 3 H₂ molecules per Mn.”

To “For H₂, a sequential pore-filling process is observed: initially, H₂ molecules exclusively occupy Site 1 (smaller pockets) at low loadings. Occupation of Site 2 (channels) commences only after reaching 8 mmol/g (~1.5 H₂/Mn), with adsorption continuing until near-saturation at approximately 3 H₂ molecules per Mn.”

Reviewer #3 (Remarks to the Author):

The manuscript by Zhang et al. presents a thorough investigation into the hydrogen isotope separation of the triazolate-based MOF [Mn(ta)₂]. The authors use advanced

experimental techniques, including in situ neutron scattering and thermal desorption spectroscopy, alongside simulations as DFT to clarify the structural dynamics and adsorption behavior of H₂ and D₂ within this framework. The reported D₂/H₂ selectivity of 32.5 at 60 K, along with the impressive enrichment of D₂ to 75% from a 5:95 mixture in just one cycle, shows that this material is a very promising candidate for large-scale deuterium separation.

It focuses on how isotopologues affects the structure, particularly the small differences in lattice expansion between H₂ and D₂, providing important insights into the separation process. The crystallographic analysis is robust, with high-quality NPD data from several facilities showing clear evidence of two distinct adsorption sites: the pockets and the channels. The time-resolved NPD experiments capture the time- and temperature-dependent adsorption behavior, revealing interesting differences in H₂ and D₂ site occupancy at 60 K. The manuscript is well-written, logically organized, and effectively placed within the broader field of MOF-based gas separation.

However, I need to address a few points that would strengthen the manuscript and enhance its clarity for a broad readership:

Question 1: The author refers to Ref. 22 for the material, but in that paper, three adsorption sites are reported, while this manuscript mentions only two. Could the authors clarify the reason for this discrepancy?

Answer: We thank the referee for highlighting this important point. In Ref. 22, He et al. defined three distinct sites in M(ta)₂: two sites in the channels connecting adjacent cavities, plus a site in the pockets. In our study of with H₂/D₂, we observed that the two channel sites from Ref. 22 exhibit nearly identical interaction energies and occupancy behaviors with hydrogen isotopes (Site 2). Our neutron diffraction data showed two clearly distinguishable regions of significant occupancy: one in the pockets (Site 1) and one in the channels (Site 2). The subtle distinctions between channel sites that might be relevant for gases with stronger quadrupole moments (like CO₂ in Ref. 22) become less significant for the nearly spherical potential of H₂/D₂ molecules, leading us to define them as a single adsorption site.

Question 2: Another related question concerns the adsorption isotherms. The Ar/N₂ isotherms exhibit a two-step behavior, which typically indicates the presence of two distinct sites. However, the H₂/D₂ isotherms show only one step. Could the authors explain the source of this difference?

Answer: This discrepancy is primarily due to the different kinetic diameters of the molecules (N₂: 3.64 Å, Ar: 3.40 Å versus H₂: 2.89 Å, D₂: 2.89 Å). This size difference is critical in relation to the framework's structural features, particularly the entrance apertures to the pocket sites (Site 1) which measure approximately 2.4 – 2.6 Å in diameter. For N₂ and Ar, accessing these pockets requires overcoming a significant

energy barrier, leading to sequential filling—first of the more accessible channel sites (Site 2), followed by the pocket sites (Site 1) at higher pressures once sufficient driving force exists. In contrast, the smaller H₂ and D₂ molecules encounter lower energy barriers for pocket access, allowing more concurrent filling of both sites.

Question 3: The authors observe different lattice expansions between H₂ and D₂ adsorption (1.2% vs. 1.0%). Could the authors provide clarification on how this specific structural response contributes to the enhanced separation?

Answer: The lattice expansion upon H₂ and D₂ loading can be mainly attributed to an elongation in the manganese-nitrogen bond length, while the bond lengths within the triazole rings remain unchanged. These Mn-N structural units are near the entrance of the pockets, therefore site 1 is altered more than the channel site 2. The smaller expansion for D₂ is mainly caused by a combination of the stronger framework-D₂ interaction and the smaller de Broglie wavelength of D₂. This difference in lattice expansions between H₂ and D₂ adsorption (1.2% vs. 1.0%) causes a preferential occupation of site 1 by D₂ and the high selectivity at 60 K. This is confirmed by the sequential gas loading (first H₂ then D₂) and the exchange of H₂ by D₂ at site 1 observed by NPD.

Question 4: The authors mention the possibility of tuning the pore size by using different metal ions. It would be beneficial to include a brief discussion on how such substitutions might influence the isotope selectivity.

Answer: We appreciate this suggestion. We have indeed synthesized and tested several metal-substituted variants of the [M(ta)₂] framework, using Zn²⁺, Cu²⁺ and Ni²⁺, and observed significant differences in their isotope separation performance. These findings are part of our ongoing research, and we plan to report comprehensive results in a future publication.

Question 5: The Figure 3 caption should clarify the symbols/colors used in panels c and f to distinguish between H₂ and D₂ data.

Answer: We have revised the caption for Figure 3 to clearly explain all symbols and colors used.

“Fig. 3. **Gas induced site occupancy and framework expansion.** (a-f) Data collected at one-minute intervals during isothermal gas adsorption and desorption. Top row (a-c): Results at 30 K; Bottom row (d-f): Results at 60 K. Left column (a,d): Occupation of H₂ at Site 1 (blue, pockets) and Site 2 (red, channels) as a function of total H₂ uptake. Middle column (b,e): Occupation of D₂ at Site 1 (blue, pockets) and Site 2 (red, channels) as a function of total D₂ uptake. Right column (c,f): Unit cell volume expansion of [Mn(ta)₂] as a function of total gas uptake for H₂ (purple) and D₂ (light blue). Error bars are smaller than the symbol size.”

Question 6: In the NPD experiments, the authors said that "each site, with a multiplicity of 192, can accommodate a maximum of 32 D₂/H₂ molecules per unit cell." The relationship between the multiplicity and the maximum accommodation should be clarified, as the connection between these two numbers is not immediately clear, at least for me.

Answer: We highly appreciate that the referee brought this issue to our notice. The sites for the highly disordered H₂ and D₂ molecules refine to (0.499 -0.001 0.251) and (-0.05 -0.04 -0.07). They are, therefore, very close to the crystallographic sites (1/2 0 1/4) and (-0.05 -0.05 -0.05), both with site occupancy of 32, with a maximum occupancy of 32 H₂ (D₂) molecules or 64 H (D) atoms.

Due to disorder, two general positions are refined, both with site occupancy of 192. The relative occupancy of these two sites refines (freely) to a maximum of 1/3. $192 * 1/3 = 64$ H (D) atoms or 32 H₂(D₂) molecules.

Question 7: The Methods section mentions the use of TOPAS7 and GSAS-II for Rietveld refinement but does not specify how the authors addressed potential systematic errors, such as preferred orientation or instrumental broadening. Including a sentence on this would strengthen confidence in the crystallographic analysis.

Answer: Preferred orientation in the cubic crystal system usually does not play a role but was checked anyway. There was no significant improvement in the fit using symmetry adapted spherical harmonics of low (2, 4) order. The peak broadening was successfully refined by mathematical convolution of: (wavelength as delta function * constant rectangular(hat) function * Thompson-Cox-Hastings-Z pseudo Voigt function with U, V, W and Y parameter).

Question 8: For practical applications, the stability of the framework over multiple adsorption-desorption cycles is crucial. While the manuscript focuses on single-cycle performance, it would be helpful to include some discussion on the cyclic stability of the framework.

Answer: We appreciate the reviewer's emphasis on cycling stability, which is indeed crucial for practical applications. Throughout our study, we subjected a single Mn(ta)₂ sample to more than 30 consecutive isotope separation cycles. Remarkably, we observed no performance degradation over these multiple cycles, with the initial and final measurements yielding nearly identical D₂/H₂ selectivity values (32.5 and 32.3, respectively). This consistent performance across numerous cycles demonstrates the exceptional structural and functional durability of the framework under repeated adsorption-desorption conditions.

Overall, this is a strong manuscript that would improve our understanding of MOFs for hydrogen isotope separation, supported by solid experimental and computational

evidence. The requested revisions are meant to expand the context for general readership.

We appreciate the referee's positive comments and insightful suggestions.

Changes made:

- In revised manuscript at page 9, the caption of Fig. 3:

From "Occupation at Site 1 (blue) and Site 2 (red) of a), d) H₂ and b), e) D₂ in, and c), f) the unit cell volume of, [Mn(ta)₂] as a function of gas uptake at a), b), and c) 30 K, and d), e), and f) 60 K. Each point is determined from a minute of data collected consecutively during isothermal gas adsorption and desorption. Errors are smaller than the symbol size."

To "Gas induced site occupancy and framework expansion. (a-f) Data collected at one-minute intervals during isothermal gas adsorption and desorption. Top row (a-c): Results at 30 K; Bottom row (d-f): Results at 60 K. Left column (a,d): Occupation of H₂ at Site 1 (blue, pockets) and Site 2 (red, channels) as a function of total H₂ uptake. Middle column (b,e): Occupation of D₂ at Site 1 (blue, pockets) and Site 2 (red, channels) as a function of total D₂ uptake. Right column (c,f): Unit cell volume expansion of [Mn(ta)₂] as a function of total gas uptake for H₂ (purple) and D₂ (light blue). Error bars are smaller than the symbol size"